# Apatite Stimulates the Deposition of Glomalin-Related Soil Protein in a Lowbush Blueberry Commercial Field

**Maxime C. Paré [1,\*], Pierre-Luc Chagnon [2], Joanne Plourde [1]**  **and Valérie Legendre-Guillemin [3]**

1   Laboratoire D'écologie Végétale et Animale, Département des Sciences Fondamentales, Université du Québec à Chicoutimi, Saguenay, QC G7H 2B1, Canada; joanne.plourde1@uqac.ca

2   Département des Sciences Biologiques, Université de Montréal, Montréal, QC H1X 2B2, Canada; pierre-luc.chagnon@umontreal.ca

3   Département des Sciences Fondamentales, Université du Québec à Chicoutimi, Saguenay, QC G7H 2B1, Canada; Valerie_Legendre-Guillemin@uqac.ca

\*   Correspondence: maxime.pare@uqac.ca; Tel.: +1-(418)-545-5011 (ext. 5071)

**Abstract:** Many wind-sensitive and unproductive soils could benefit from increased glomalin-related soil protein (GRSP), an operationally defined soil protein pool known to improve soil quality and nutrient storage. We expect at least part of this GRSP fraction to originate from fungal biomass. Although P-rich minerals such as apatite are known to increase C allocation from plants to mycorrhizal fungi, there are no studies directly linking apatite with GRSP. We investigated the effect of apatite on GRSP deposition rates in a cultivated field of wild lowbush blueberry (*Vaccinium angustifolium* Aiton; *Vaccinium myrtilloides* Michx.) in the Saguenay-Lac-Saint-Jean region of Quebec (Canada). A field incubation technique (145 days) using sterilized porous sand bags (50 μm pores) was used to measure in situ easily extractable GRSP (EE-GRSP) deposition rates from bags with ($n = 10$) and without ($n = 10$) apatite. Half of the bags ($n = 10$) were also soaked in Proline®480 SC (Bayer CropScience, Calgary, Alberta, Canada) (Prothioconazole) to determine if EE-GRSP deposition rates were affected by this commonly applied fungicide. Our results indicated that adding apatite into sand bags significantly increased (+70%) EE-GRSP deposition rates, whereas soaking the bags in fungicide had no significant effect. Although the direct linkage between GRSP and lowbush blueberry plants remains to be detailed, our study reports for the first time GRSP concentrations from lowbush blueberry soils. Implications of these findings are discussed.

**Keywords:** *Vaccinium angustifolium*; *Vaccinium myrtilloides*; apatite; wild blueberry; glomalin

## 1. Introduction

Canada is the world's leading producer of wild lowbush blueberry (*Vaccinium angustifolium* Aiton; *Vaccinium myrtilloides* Michx.) producing about 60,000 tons of fruit per year [1]. The northern limit for agriculture in the Saguenay-Lac-Saint-Jean (SLSJ) region of Quebec, Canada, is the most important area for this crop with about 30,000 ha in production [1]. Existing commercial blueberry fields operate on poorly structured, acid, and wind-sensitive sandy soils, such as the L'Afrique and Parent soil series [2]. In the SLSJ region, the use of mechanical- and/or burn-pruning coupled with a shallow organic matter (OM) layer overtopping wind-sensitive sands can result in the development of unproductive areas, marked by bare soils lacking both OM and plants. These bare soils—often located at the top of sand dunes—may sometimes represent 50% of the landscape [3].

Soil glomalin content is a valuable and reliable indicator of soil health [4]. Glomalin has been associated with increased soil water aggregate stability [5–9] as well as soil organic carbon

(C) and nitrogen (N) content [4,10–13]. However, there has been some controversy regarding the composition and the origin of soil glomalin. Glomalin was thought to be a glycoprotein originating from arbuscular mycorrhizal (AM) fungi [14] and reaching soil through AM hyphal necromass [15]. However, several lines of evidence have brought a more nuanced definition and questions regarding even the glycoprotein nature of glomalin. For instance, glomalin was found to be co-extracted with humic substances [16–18] and, therefore, has a spectral signature more related to humic materials than to carbohydrates [19]. Based on this, Rillig [20] suggested that part of this operational pool of glomalin [termed as glomalin-related soil proteins (GRSP) and simply defined as any substance extracted by autoclaving soil in a citrate buffer and quantified using a colorimetric protein assay] may also come from general fungal hydrophobins (i.e., not necessarily of AM origin). This new definition of glomalin agrees with recent studies that show substantial amounts of glomalin retrieved in non-mycorrhizal roots [21] and in soils supporting stands dominated by ectomycorrhizal hosts such as beech, fir, and hemlock [13].

Following the idea that glomalin may originate from numerous paths and organisms, GRSP deposition may occur in infertile and wind-sensitive wild blueberry fields that are largely dominated by ericaceous shrubs (*Vaccinium* and *Kalmia* can comprise about 95% of the vegetation of sand dunes), which are not AM plants (although there are sparse records of AM colonization in ericaceous hosts in the literature [22–25]). Given the well-recognized importance of GRSP to soil fertility, irrespective of its origin [19], this operational pool could represent a useful biochemical indicator of soil fertility and quality [26–28]. Therefore, the challenge may not necessarily be to identify mechanistically the origin of GRSP, but rather to identify the drivers of its production.

To better rehabilitate wind-sensitive soils, such as those found in SLSJ wild blueberry fields, management practices that may influence GRSP should be investigated. More specifically, it would be useful to identify management practices that are most likely to increase and/or decrease GRSP deposition rates. Given the presumed fungal contribution to soil GRSP pool [7,20], a better understanding of the factors promoting soil hyphal foraging is required. For example, wild blueberry management practices related to phosphorus (P) fertilization and fungal disease (e.g., fungicides) should be investigated.

Rocks that are rich in P, such as apatite [$Ca_5(PO_4)_3(OH, F, Cl)$], contain about 15%–18% of water-insoluble P fractions (34–42% of $P_2O_5$); however, this P is not available for plant uptake [29,30]. To access this soil P resource, the majority of plants have evolved a symbiotic relationship with fungi [31]. Mycorrhizal fungi can extract and obtain—helped by bacteria and other fungi—P from apatite and then translocate a portion of this P to the host plant [32,33]. Adding apatite to P-depleted environments, such as wild blueberry soils [34,35], could increase C allocation to mycorrhizal symbionts, in return for apatite-derived P [36]. Results from P-poor soils (Podzols) in Sweden indicate that host plants enhanced C allocation to their mycorrhizal symbionts when the latter colonized apatite [37]. However, no agricultural study has yet directly linked the application of apatite to GRSP deposition.

Here, we investigate the effect of apatite, a P-rich mineral, on GRSP deposition rates in a commercial blueberry field in the Saguenay-Lac-Saint-Jean (SLSJ) region. Furthermore, as many landowners use fungicides to treat their crops against fungal pathogens (e.g., *Botrytis cinerea*, *Monilinia vaccinii-corymbosi*, and *Valdensia heterodoxa*), we also investigate the effects of a commonly used fungicide on GRSP deposition rates. We hypothesize that apatite would stimulate GRSP deposition rates, whereas the fungicide would inhibit GRSP deposition.

## 2. Materials and Methods

A field soil incubation protocol was adapted from Lovelock et al. [28] and Berner et al. [37] to quantify GRSP deposition rates. The experimental approach is based on incubating porous nylon bags that are filled with sterile and OM–free sand. At the end of the experiment, the amounts of GRSP in the sand bags are measured to determine the amount deposited during the field incubation period.

### 2.1. Sand Bag Preparation

A sample of local sandy soil was collected manually and placed in a muffle furnace at 550 °C for 48 h to sterilize the soil and remove all OM [38]. Sterilized and OM–free soil was then sieved using 40 and 100 mesh sieves; only sand particles between 150 and 425 μm were used to fill the bags.

About 60 g of sterilized and sieved sand was poured into 8 × 12 cm nylon bags (SEFAR NITEX, Filmar, Montreal, Quebec, QC, Canada) having 50 μm pores (Figure 1). This pore size was selected because it (1) minimizes sand contamination (inputs) and losses (outputs) during incubation and (2) allows soil microorganisms (fungal hyphae and bacteria) to colonize the sand bag [39], as GSRP is likely produced by those soil microorganisms [19]. Waterproof glue (LePage®), Mississauga, Ontario, Canada) was used to seal the bags (Figure 1). To test our hypothesis, we applied two treatments to the bags: (1) apatite (with and without apatite) and (2) fungicide (with and without fungicide) in a complete factorial design having five replications. In total, 20 experimental units were used (2 apatite × 2 fungicide × 5 replicates).

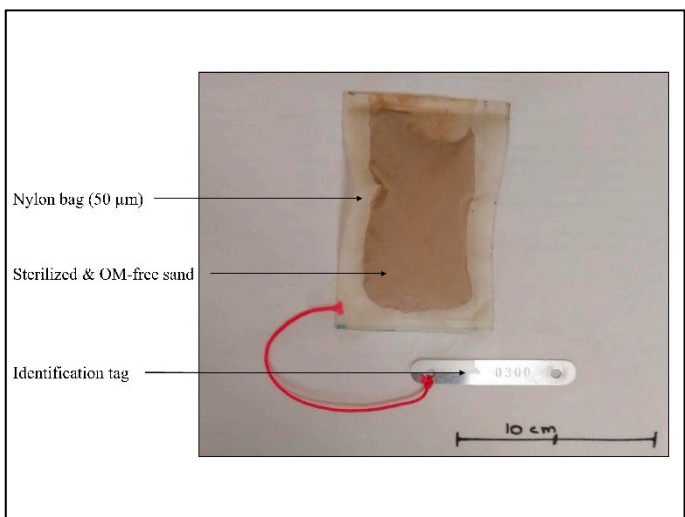

**Figure 1.** An example of a nylon sand bag. The bag (50 μm pores) is filled with sterilized and organic matter (OM)-free sand.

For the bags with apatite (*n* = 10), 0.10 g (about 5 g of 2 mm in diameter) of hydroxyapatite [$Ca_5(PO_4)_3(OH)$] was randomly and uniformly incorporated within the sand. Analysis by laser ablation (M-50 Excimer, Australian Scientific Instruments, Fyshwick, Australia)–inductively coupled plasma–mass spectrometer (LA-ICP-MS) (7700 x, Agilent, Santa Clara, CA, USA) showed that apatite contained about 37.8% ± 0.9% of $P_2O_5$ and 37.7% ± 0.6% of calcium (Ca). The filled sand bags were then sealed using the glue. The sand bags with fungicide (*n* = 10) were soaked 2 min in a solution containing 4.2 mL·L$^{-1}$ of a broad-spectrum systemic commercial fungicide (Prolines 480 SC, active ingredient: Prothioconazole). This concentration of fungicide represents the mixture that is normally applied by blueberry producers [40]. The other bags not treated with fungicide (*n* = 10) were soaked in deionized water. All bags were then air-dried at 25 °C for 2 days before field incubation.

### 2.2. Field Soil Incubation

The sand bags were placed within a commercial lowbush blueberry field near the town of Albanel (48°54′22″ N; 72°19′57″ W) in the SLSJ region. Although the producer applies 20 kg of $P_2O_5$ ha$^{-1}$ every two years, the selected field was extremely poor in P, reflected by low P saturation index (Table 1). The bags were placed in a 5 × 4, equally spaced grid design (20 plots, spaced 5 m apart) on a homogeneous and flat section of the field. Plant cover was dominated by lowbush blueberry (90%) with the remaining plant cover (10%) being mostly sheep laurel (*Kalmia angustifolia* L.) and Canadian dwarf cornel (*Cornus canadensis* L.). On 18 May 2016 one sand bag per plot (randomly attributed) was

gently inserted horizontally (to minimize disturbance) into the F soil horizon (2–5 cm) using a hand trowel. The mineral soil under organic layers (>5 cm) was a loamy sand (LS) with about 80%–85% of sands. All bags (*n* = 20) were collected after 145 days of field soil incubation (10 October 2016). The sand inside each bag was gently recovered to minimize contamination, then homogenized, and kept cool (4 °C) until GRSP extraction.

**Table 1.** Initial chemical properties of the mineral soil (Mistassini loamy sand) used for this study.

| Soil Properties | Method | Value |
|---|---|---|
| Soil pH | In water (1:1) | 4.4 |
| Soil organic matter (%) | Combustion | 5.1 |
| P (mg·kg$^{-1}$) | Mehlich 3-Extractable | 37.1 |
| Al (mg·kg$^{-1}$) | Mehlich 3-Extractable | 1864 |
| P saturation index (%) | (P/Al)×100 | 2.0 |
| K (mg·kg$^{-1}$) | Mehlich 3-Extractable | 21.4 |
| Mg (mg·kg$^{-1}$) | Mehlich 3-Extractable | 9.8 |
| Ca (mg·kg$^{-1}$) | Mehlich 3-Extractable | 121.0 |

*2.3. Glomalin-Related Soil Protein (GRSP) Extraction and Quantification*

Extraction and quantification of the easily extractable GRSP (EE-GRSP) followed the protocol of Reyna and Wall [41]. EE-GRSP is considered to be the most recently deposited fraction of glomalin [5]. For each sample, 1 g of equivalent dry sand was put into a 50 mL centrifuge tube. The sand was then mixed with 8 mL of sodium citrate solution (20 mM) at pH 7. Tubes were autoclaved at 121 °C for 30 min (SI-120 Scientific Isothermal Sterilizer, Steris, Quebec City, Quebec, Canada) and then immediately centrifuged (Avanti J-E, Beckman Coulter, Brea, CA, United States) at 5000 g for 15 min. The supernatant was collected and kept frozen until the EE-GRSP was quantified.

Quantification of EE-GRSP was performed by colorimetry using a BCA assay [42]. All samples were quantified in triplicate. Samples were placed in a 96-well flat-bottomed microplate. For each well, 25 μL of supernatant containing EE-GRSP was mixed with 200 μL of standard working reagent. Bovine serum albumin (BSA) was used as a standard. Microplates were incubated at 37 °C for 2 h; the absorbance was then read at 562 nm with a microplate spectrophotometer set at 27 °C (Multiskan Go, Thermo Scientific, Waltham, MA, USA). Extraction and quantification protocols were also applied to the initial sterilized and OM–free sand to benchmark the incubation and laboratory procedures. No EE-GRSP was detected (0 mg of EE-GRSP kg$^{-1}$ of sand$_{dry}$) from these control samples.

*2.4. Statistical Analysis*

Deposition rates of EE-GRSP (mg of EE-GRSP kg$^{-1}$ of sand$_{dry}$ day$^{-1}$) were calculated by dividing the concentration of measured EE-GRSP (mg kg$^{-1}$) at the end of field incubation per 145 days (the duration of the field incubation). A linear model (Univariate procedure in SPSS) was used for variance analysis. Data normality (Kolmogorov–Smirvov's test) and homoscedasticity (Levene's test) of the variance were verified before any analyses. All statistical analyses were performed using SPSS, version 24 for Windows [43] with a significant level of 0.05.

**3. Results**

Adding small amounts of apatite into sand bags (1 mg of apatite per g of sand) increased significantly the EE-GRSP deposition rates by 70% (+1.66 mg kg$^{-1}$ day$^{-1}$), whereas soaking the bags in fungicide did not significantly influence EE-GRSP deposition rates (Table 2, Figure 2). A non-significant interaction found between apatite and fungicide treatments suggests that apatite independently increased EE-GRSP deposition rates, regardless of the applied fungicide treatment.

**Table 2.** F-values and probabilities obtained from a generalized linear model to test the effects of apatite and fungicide on easily extractable glomalin-related soil protein (EE-GRSP) deposition rates.

| Source of Variation | df | SS | *F*-Value | Probability |
|---|---|---|---|---|
| Apatite (w/ vs. w/o apatite) | 1 | 0.338 | 5.10 | 0.038 |
| Fungicide (w/ vs. w/o fungicide) | 1 | 0.000 | 0.01 | 0.941 |
| Apatite × Fungicide | 1 | 0.096 | 1.45 | 0.246 |
| Error | 16 | 1.061 | | |
| Total | 19 | | | |

df: degrees of freedom; SS: sum of squares (Type III); w/: with, w/o: without.

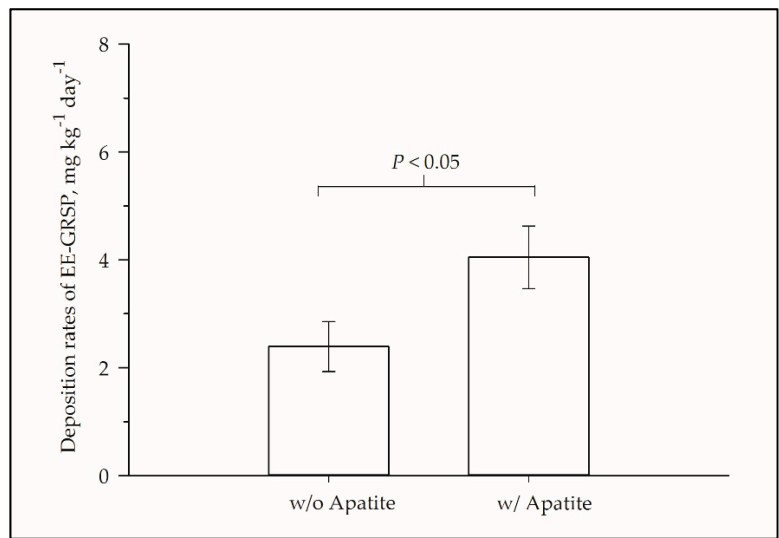

**Figure 2.** Easily extractable glomalin-related soil protein (EE-GRSP) deposition rates (per day) in sand bags. Two groups of sand bags were defined based on whether they contained (w/) or lacked (w/o) apatite. Error bars represent the standard error (SE) from the mean ($n = 10$). Averages are significantly different at $P < 0.05$ level.

## 4. Discussion

Our study reports, for the first time, glomalin deposition rates from a lowbush wild blueberry field. Results are consistent with other findings [13] where significant rates of GRSP deposition were found even in stands not dominated by AM fungal hosts. The lack of a significant effect of fungicide (Table 2) may be explained by the resiliency of the fungal community to a single application of the fungicide, as many studies have shown the rapid recovery and recolonization of soil fungal communities after applying fungicide [44]. Our sand bag experiment demonstrates that GRSP can accumulate in a root-free environment (i.e., sand bags) and that GRSP has a fungal and/or bacterial (microorganisms growing near/on these hyphae) origin [45]. As such, an important research avenue will be to further improve our understanding of the factors favoring fungal foraging in the soil and, therefore, likely maximize GRSP deposition rates.

The positive effect of apatite on GRSP (Figure 2) is consistent with previous published findings. For example, the addition of apatite into incubated sand bags within various P-depleted Norway spruce (*Picea abies* L.) forest soils, was also associated with an increase of fungal biomass [37,46]. It has also been demonstrated that host plants enhance C allocations to their mycorrhizal symbionts colonizing apatite [37]. Bidartondo et al. [47] found a 2.5-fold increase in the growth of extrametrical mycelium of some ectomycorrhizal fungal species after the addition of apatite. However, it has been suggested that a soil needs to be depleted in plant-available P to stimulate fungal foraging [48]. Apatite added to sand bags that were inoculated within a Norway spruce soil, did increase mycorrhizal fungal mycelial growth in the N-treated samples, whereas mycelial growth was reduced by about 50% when N was added alone [49]; as authors suggested, adding N likely created a P-depleted soil environment for

the plants. Adding apatite to a P-depleted soil environment (from a plant's perspective) is known to stimulate mycorrhizal symbionts in soils but, to our knowledge, the relationship between apatite and GRSP has never been shown directly. Nevertheless, a controlled environment experiment by Smits et al. [50], using radiocarbon ($^{14}$C), demonstrated that under P-limitation, plant C allocation to fungi increased by 17 times in the surrounding area where apatite was added. However, it was not determined if this increase in C allocation was also reflected in GRSP content.

By improving soil quality (e.g., enhanced soil structure and aggregation) through increased GRSP content, the application of apatite could eventually lead wild blueberry producers to improve soils affected by wind erosion (currently unproductive areas). A study using soil samples collected from various sites throughout the United States that were subjected to different soil management practices found a strong positive relationship between EE-GSRP and soil aggregate stability to water [5]. Rillig et al. [51] also showed that the direct effect of GRSP on aggregate stability was greater than the effect of fungal hyphae alone.

The role of GRSP in improving soil C storage has been well documented [4], as GRSP can represent a significant proportion of OM, reaching as high as 5% of total C in soils [52]. Two main reasons can explain this. First, the C recalcitrance of GRSP slows down soil C turnover rates, and this recalcitrance chemically protects GRSP from heterotrophic decomposition [53]. Second, its hydrophobic and binding nature stabilizes soil aggregates and other soil C molecules that are trapped in aggregates and, therefore, physically protects organic C from soil decomposers [54]. Therefore, our results suggest that the application of apatite could eventually increase soil C storage through GRSP deposition. However, we did not model soil C storage related to GRSP deposition because young fungal communities require more C allocation than does an established community [55]. Thus, there remains a need to assess GRSP deposition through long-term field studies before offering any recommendations to stakeholders.

Our results should promote applied research projects focused on improving soil quality. The addition of apatite to P-poor soils increases GRSP deposition. In turn, these changes favor enhanced soil structure and soil particle aggregation as well as greater C storage. Therefore, coupled with current practices (e.g., applying and leaving wood straw at the soil surface), the application of apatite could eventually lead blueberry landowners to establish new plant-productive lands from currently unproductive areas through increased GRSP deposition.

## 5. Conclusions

Our study showed that adding apatite increased EE-GRSP deposition rates in a commercial blueberry field, whereas soaking sand bags in fungicide did not influence EE-GRSP deposition rates. A better understanding of the mechanisms and processes involved in generating EE-GRSP is critical before prescribing any recommendations to producers. First, direct links between blueberry plants, ericoid mycorrhizal symbionts, and GRSP production remain to be identified. Without this information, adding apatite could also benefit and favor other plants considered as weeds to wild blueberry producers. Second, it is critical to understand any temporal variations in GRSP deposition rates, since fungal colonization of sand bags may peak during the autumn in boreal ecosystems [39]. Moreover, it is essential to monitor glomalin deposition rates in other types of blueberry fields (e.g. old vs. recently established fields) as young fungal communities require a greater C allocation than do older ones [55]. Other apatite-rich residues may also be tested as sources for increasing GRSP content. For example, apatite-rich tailings from niobium extraction in the SLSJ area could represent an interesting alternative to the more expensive apatite rocks. These results open many new research opportunities for soil science and land conservation.

**Author Contributions:** The paper is the result of the collaboration among all authors. Conceptualization, M.C.P and J.P.; methodology, J.P. and M.C.P.; validation, M.C.P. and V.L.-G.; formal analysis, M.C.P.; investigation, M.C.P.; resources, M.C.P.; writing—original draft preparation, M.C.P.; writing—review and editing, P.-L.C. and V.L.-G.; visualization, M.C.P.; supervision, M.C.P. and V.L.-G.; project administration, M.C.P.; funding acquisition, M.C.P.

**Funding:** This study was funded by the Fonds de développement de l'Université du Québec à Chicoutimi (FUQAC).

**Acknowledgments:** The authors thank M. Plourde (Bleuet Royal) for giving access to his blueberry field as well as Dr Paul Bédard and Dre Annick Doucet for providing the apatite and fungicide, respectively. Thanks also to Dany Savard (LabMaTer), Andréanne Simard, Catherine Tremblay, Claire Fournier, and Caroline Côté for their laboratory and technical assistance. We also want to thank the anonymous reviewers for their careful reading and their many insightful comments and suggestions.

**Conflicts of Interest:** The authors declare no conflict of interest. The funders had no role in the design of the study; in the collection, analyses, or interpretation of data; in the writing of the manuscript, or in the decision to publish the results.

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
