# Peer review of "Apatite Stimulates the Deposition of Glomalin-Related Soil Protein in a Lowbush Blueberry Commercial Field"

_agriculture, doi:10.3390/agriculture9030052_

Reviewer 1 Report

Results: Could include results from soil structure and soil particle aggregation as well as greater C storage changes. These results could help to better understand the process of improving soil quality

 Discussion

In the discussion it must relate the results of the characteristics of the soil, for example the PH. Soil pH is acid PH (4.4.), but which will be the production of Glomalin to PHS alkaline. Also, this discussion could be be done to relate the organic matter and the phosphorus content in the soil.

 Author Response

See attached file for responses.

Reviewer 2 Report

Comments               

I appreciated the opportunity to review the manuscript entitled ( Apatite stimulates the deposition of glomalin-related3 soil protein in a lowbush blueberry commercial field). This study aimed to investigate the effect of apatite on GRSP deposition rates in a cultivated field of wild lowbush blueberry (Vaccinium    angustifolium Aiton; Vaccinium myrtilloides Michx.) in the Saguenay‒Lac-Saint- Jean region of Quebec (Canada).

 I have noticed the following points:

1-   There is an obvious lack of the data provided, thus it would be better to be considered as a short communication paper.

2-   In the abstract, you should abstain using the personal pronouns in the whole paper.

3-   In line 32-33, you reported that Canada is the world’s leading producer of wild lowbush blueberry fruits (Vaccinium angustifolium Aiton; Vaccinium myrtilloides). This sentence must be associated to reference.   

4-   Line 40 dunes—may sometimes represent 50% of the total surface in production [3]. An Obvious weakness in English language. Clear such and reorganize.

5-   Line 66, use to identify instead of identify only.

6-   Line 86, you stated the hypothesis at the end of the introduction after the aims, and that is a bit unusual. It must be before aims.

7-   Line 93-94, how did you make sure that all O.M has been removed? And why did you choose the exact temperature (550) without referring to any reference?

8-   Line 128, what is the soil texture of the field soil and what are the soil particles proportions? Must be reported here.

9-    Line 149, not understandable how worked out the deposition rate of GRSP? How did you use the days to get such rate? Would you please clarify it more?

10-                      Line 153, statistically, the less significant differences for this experiment (P value p< 0.05) was not mentioned. Must be there.

11-                      Line 114-115, the concentrations of fungi acid used here seem not to be the right based on the data shown as no effect on glomalin rates was found. It should apply fungi acid several times to control and remove all fungal biomass until the experiment lined perfectly.

12-                      Line 195-201, this paragraph looks like a review rather than a real discussion. Moreover, you fingered so many times soil aggregations and soil structure but this parameter was not taken into account in this study. You should get rid of this one and focus on your measured parameter (apatite and glomalin).

13-                      The whole discussion is become like a review instead of a deep discussion.  Providing your scientific opinion is under priority. Neglecting such will weak your discussion.

14-                      Line 229, giving a reference in the conclusion section is weird, and must be removed out of the section. 

Author Response

See attached file for responses.

Round  2

Reviewer 1 Report

They responded adequately to the observations. I agree with the change to short communication